# Advancing Internal Dosimetry in Personalized Nuclear Medicine: Toward Optimized Radiopharmaceutical Use in Clinical Practice

**DOI:** 10.3390/ph18111741

**Published:** 2025-11-17

**Authors:** Ali H. D. Alshehri

**Affiliations:** Department of Radiological Sciences, College of Applied Medical Sciences, Najran University, Najran 61441, Saudi Arabia; ahzafer@nu.edu.sa

**Keywords:** Monte Carlo simulation, GATE and GAMOS codes, MIRD formalism, radiopharmaceutical dosimetry, Zubal phantom, ^99m^Tc

## Abstract

**Background:** Quantifying absorbed doses from radiopharmaceuticals within human organs necessitates advanced computational modeling, as direct in vivo measurement remains impractical. **Methods:** In this study, three Monte Carlo-based simulation codes, Monte Carlo N-Particle version 6 (MCNP6), GEANT4 Application for Tomographic Emission (GATE), and GEANT4-based Architecture for Medicine-Oriented Simulations (GAMOS), were employed to evaluate internal dosimetry following the Medical Internal Radiation Dose (MIRD) formalism. As an illustrative case, simulations were first performed for ^99m^Tc-MIBI uptake in the myocardium using the anthropomorphic phantom, with the heart modeled as the source organ to assess energy deposition in key target organs. Dose assessments were conducted at two time points: immediately post-injection and at 60 min post-injection (representing the cardiac rest phase), allowing comparison against established clinical reference data. **Results:** Across all codes, organ-specific dose distributions exhibited strong consistency. The pancreas absorbed the highest dose (GATE: 21%, GAMOS: 20%, MCNP6: 22%), followed by the gallbladder (GATE: 18%, GAMOS: 17%, MCNP6: 18%) and kidneys (GATE: 16%, GAMOS: 15%, MCNP6: 16%). These findings established a consistent organ dose ranking: pancreas *>* gallbladder *>* kidneys *>* spleen *>* heart/liver, corroborating previously published empirical data. To demonstrate the versatility of the framework, additional simulations were performed with ^18^F in an anthropomorphic phantom and with spherical tumor models using therapeutic radionuclides (^177^Lu and ^225^Ac). This broader application underscores the adaptability of the tri-code approach for both diagnostic and therapeutic scenarios. **Conclusions:** This comparative analysis highlights the complementary advantages of each Monte Carlo platform. GATE is well-suited for high-fidelity clinical applications where anatomical and physical accuracy are critical. GAMOS proves advantageous for rapid prototyping and iterative modeling workflows. MCNP6 remains a reliable benchmark tool, particularly effective in scenarios requiring robust radiation transport validation. Together, these Monte Carlo frameworks form a validated and adaptable toolkit for advancing internal dosimetry in personalized nuclear medicine, supporting both clinical decision-making and the development of safer, more effective radiopharmaceutical therapies.

## 1. Introduction

Nuclear medicine represents a transformative intersection of molecular biology and radiopharmaceutical science, enabling non-invasive and functional imaging and targeted therapy [1,2]. Among diagnostic radionuclides, technetium-99m (^99m^Tc) remains the most widely used, owing to its favorable physical characteristics: a short half-life of 6.02 h, a monoenergetic 140 keV gamma emission, and rapid systemic clearance [3,4]. These characteristics minimize patient radiation burden while providing high-resolution visualization of physiological processes such as myocardial perfusion, pulmonary ventilation, and thyroid function [5]. Furthermore, ^99m^Tc can form stable complexes with diverse ligands, including sestamibi for cardiac imaging and macroaggregated albumin for pulmonary scans, or be administered as sodium pertechnetate for thyroid and gastrointestinal evaluations [5].

Accurate estimation of absorbed dose remains central to the clinical utility of nuclear medicine. Unlike external beam radiotherapy, where dose delivery can be tightly conformed [6], internal dosimetry must account for heterogeneous tracer distribution, dynamic biokinetics, and patient-specific anatomy [7,8]. Direct in vivo dose measurement is not feasible, necessitating advanced computational models to estimate organ-level radiation exposure and mitigate risks of toxicity or secondary malignancies. The Medical Internal Radiation Dose (MIRD) formalism, developed by the Society of Nuclear Medicine, provides standardized dosimetric quantities, such as the specific absorbed fraction (SAF) and the S-value [9,10,11]. The SAF quantifies the fraction of emitted energy absorbed per unit mass in a target organ from a source organ, while the S-value represents the mean absorbed dose per unit accumulated activity [6]. These parameters are typically derived using voxelized anthropomorphic computational phantoms, which simulate radiation transport and energy deposition in anatomically realistic models [10]. Monte Carlo (MC) methods are widely regarded as the gold standard for internal dosimetry. General-purpose MC codes such as GATE (Geant4 Application for Tomographic Emission), GAMOS (GEANT4-Based Architecture for Medicine-Oriented Simulations), and MCNP6 (Monte Carlo N-Particle Code, Version 6) have been extensively applied in medical physics. While their individual strengths are well documented GATE for high anatomical realism and detailed particle tracking [12,13], GAMOS for computational efficiency and modularity, and MCNP6 for its established reliability in photon and neutron transport [14]. A systematic side-by-side comparison of these codes under a standardized simulation framework for internal dosimetry has not been comprehensively reported [15].

The present study addresses this gap by benchmarking MCNP6, GATE, and GAMOS for internal dosimetry calculations [16]. Using the Zubal anthropomorphic phantom, SAFs, S-values, and organ-level absorbed doses for ^99m^Tc distributions were computed at two physiological states (immediately post-injection and at 60 min post-injection, corresponding to the cardiac rest phase). Results were compared against clinical reference data from OLINDA/EXM. To illustrate the broader applicability of the framework, additional simulations with ^18^F distributions and spherical tumor models were performed. Ultimately, this study establishes a validated, multi-code computational dosimetry platform that integrates anatomical realism with methodological flexibility. Such a framework supports the evolving field of theranostics, combining diagnostic imaging and radionuclide therapy with agents like ^177^Lu-PSMA or ^225^Ac, where accurate dosimetry is essential to balance therapeutic efficacy with protection of dose-limiting organs.

## 2. Results

### 2.1. Absorbed Dose

This study aimed to quantify the absorbed dose delivered by several clinically relevant radioisotopes when uniformly distributed within a small spherical tumor volume. A 1 cm^3^ tumor model, centered within a cold (non-radioactive) water bath, was employed to represent a localized radiotracer distribution embedded in a biologically inert medium. The selected tumor size approximates the average volume for micro metastases and small lesions, acknowledging that actual tumor geometries may vary considerably. The geometric configuration utilized in the simulation setup is illustrated in Figure 1.

For each radionuclide listed in Table 1, MC simulations were performed to estimate the total absorbed dose delivered to the tumor. These values were analytically integrated over time by assuming mono-exponentially physical decay of the radionuclide. Specifically, the absorbed dose [*D*(*t* = ∞) = *D*_0_*τ*] was calculated as the product of the initial dose rate (*D*_0_) and the physical decay constant time (*τ*), given by: [*τ* = *T*_1/2_/ln(2)], where *T*_1/2_ denotes the physical half-life of the radionuclide. This approach provides an analytical integration of the dose over an infinite time period, under the idealized assumption of no biological clearance and exclusive reliance on physical decay.

Each radioisotope was characterized by its fundamental nuclear properties, including atomic number (*Z*), mass number (*A*), ionic charge (*Q*), and excitation energy (*E*), as detailed in Table 1. The simulations incorporated comprehensive decay schemes and de-excitation processes to model emission spectra and energy deposition accurately. While computationally demanding, this source definition method provides a high level of dosimetric fidelity and is essential for robust evaluations in both therapeutic and diagnostic nuclear medicine applications.

Table 2 presents a comparative analysis of absorbed dose calculations for several widely used isotopes in nuclear medicine, based on simulations conducted using GATE, GAMOS, and MCNP6 codes. Each simulation was performed for a standardized activity of 1 mCi. Across all isotopes, a high degree of consistency was observed between the three codes, with slight variations attributed to differences in their MC transport algorithms, energy cut-offs, and cross-section libraries. The highest absorbed dose was associated with the high-energy beta emitter ^90^Y, with values ranging from 1207.89 Gy (GAMOS) to 1215.34 Gy (MCNP6). For diagnostic gamma emitters like ^18^F and ^99m^Tc, the absorbed doses remained low and closely matched; for example, for ^99m^Tc: 15.22 Gy (GATE), 15.11 Gy (GAMOS), and 15.30 Gy (MCNP6).

Table 3 presents a comprehensive comparison of absorbed dose estimations for ^131^I and ^99m^Tc sources using three MC codes: GATE, GAMOS, and MCNP6. For ^131^I radiotracer simulations, the absorbed doses were consistent across all codes, with GATE yielding 952.55 Gy, GAMOS 948.60 Gy, and MCNP6 956.80 Gy for 1 mCi activity, indicating a deviation of less than 1% among them. When considering an electron source at 191 keV, a slight increase in dose was observed, with MCNP6 predicting the highest value at 1038.40 Gy. Regarding ^99m^Tc, the absorbed dose calculated for the radiotracer emission was 15.22 Gy in GATE, closely matched by GAMOS (15.10 Gy) and MCNP6 (15.35 Gy). For mono-energetic electron sources at different energies (1.6 keV to 137 keV), the dose calculations showed excellent agreement among the three codes, with variations typically within ±1–2%.

### 2.2. Zubal Phantom Study Using 18-Fluorine

In this study, voxelized sources were incorporated through two primary approaches: either by reading ASCII files or by importing InterFile images. The InterFile format was selected owing to its widespread compatibility with digital phantoms and clinical patient datasets, enabling a direct interpretation of emission distributions.

A realistic anthropomorphic phantom, based on the Zubal model, was employed as the emission source. This phantom provides anatomically accurate, voxel-based representations of the human body, capturing variations in organ size, shape, and orientation to closely mimic real human anatomy. The Zubal phantom was selected due to its established use in nuclear medicine and internal dosimetry research, where its high degree of anatomical realism and compatibility with voxel-based Monte Carlo simulations have been extensively validated. Importantly, it offers distinct activity and attenuation datasets, making it highly suitable for simultaneous emission-attenuation simulations in hybrid imaging frameworks. Each simulation generated two primary datasets: an activity phantom and an attenuation phantom. Within the activity phantom, key organs including the heart, liver, lungs, kidneys, spleen, and surrounding background tissues were explicitly modeled. The background component accounted for residual anatomical structures present within the imaging field of view. The attenuation phantom contained organ-specific attenuation properties as defined by the original phantom model. Given that computed tomography (CT) imaging fundamentally requires an attenuation map, a conversion of the attenuation dataset to the InterFile format was performed to ensure seamless integration within the simulation framework.

Subsequently, the voxelized phantom was employed to define the emission source for the simulation of radiopharmaceutical distributions under realistic conditions. A radiotracer dose of 1 mCi of ^18^F was administered individually to the heart, kidneys, and a combined heart-plus-kidney configuration. The absorbed dose results are summarized in Table 2. The spatial distribution of activity uptake within the heart and kidney regions is depicted in Figure 2, while Figure 3 present representative transaxial slices of the corresponding three-dimensional dose distributions.

Table 4 summarizes the calculated absorbed doses in the heart and kidney organs for a 1 mCi injection of ^18^F radiotracer, using GATE, GAMOS, and MCNP6 simulation codes. The simulations accounted for both standard bio-distribution and enhanced organ-specific uptake scenarios, denoted as (+), where 97% of the activity was localized in the respective target organ. For the heart without enhanced uptake, the absorbed doses were 0.5464 Gy, 0.5405 Gy, and 0.5510 Gy for GATE, GAMOS, and MCNP6, respectively. Under enhanced uptake conditions, the doses increased accordingly, with MCNP6 consistently yielding slightly higher values. Similarly, for the kidney, the absorbed dose estimates under both baseline and (+) uptake models showed strong consistency among the three codes, with discrepancies below 1–2%. When both the heart and kidney were targeted, the total absorbed doses ranged from 1.4035 Gy to 1.4180 Gy in the standard scenario, and from 1.7802 Gy to 1.7985 Gy under (+) conditions.

Figure 4 presents a comparative visualization of absorbed dose calculations for various organ configurations using the GATE, GAMOS, and MCNP6 Monte Carlo simulation codes with a fixed activity of 1 mCi of ^18^F. The data show consistent results across all three simulation platforms, with variations remaining within a narrow range (maximum deviation less than 1%).

Notably, when a 97% radiotracer uptake is applied to both the heart and kidney, a significant increase in absorbed dose is observed. For example, the dose to the heart increases from approximately 0.546 Gy to 0.884 Gy using GATE. This trend is similarly observed in the kidney dose estimations and in the combined heart + kidney scenarios, underscoring the substantial impact of biodistribution and uptake on organ dosimetry.

### 2.3. Dose Estimation Using GATE, GAMOS, and MCNP6 Codes

Monte Carlo radiation transport simulations were conducted using GATE, GAMOS, and MCNP6 to calculate the SAF and corresponding S-values (mGy/MBq·h) for ^99m^Tc gamma emissions, with the heart modeled as the primary source organ. Figure 5 illustrate the simulated ^99m^Tc activity distribution within the heart and the derived SAF values for anatomically adjacent target organs. The detailed results are presented in Table 5.

All three codes produced consistent results, with GATE providing enhanced spatial resolution due to its voxelized geometry, whereas GAMOS and MCNP6 yielded comparable estimates with minor numerical deviations. These variations are attributed to the different geometry handling and physics modeling capabilities of each simulation engine. MCNP6, in particular, was used to validate the consistency and robustness of the dosimetric results obtained using GATE and GAMOS.

Figure 6 graphically compares the S-values estimated using GATE, GAMOS, and MCNP6 for ten organs, with the heart serving as the primary source of ^99m^Tc emissions. The results exhibit high consistency among the three codes, with deviations generally below 3%, demonstrating the robustness and reproducibility of Monte Carlo-based dosimetric calculations. The self-dose to the heart was predictably the highest (3.30 × 10^−6^ mGy/MBq·h in GATE), due to the proximity of source and target voxels. Organs with smaller voxel counts and further anatomical distance, such as the brain and bladder, exhibited lower S-values.

### 2.4. Post-Injection Dynamics at Cardiac Rest

Figure 7 illustrates the SAF values for the heart and selected target organs at 60 min post-injection, as summarized in Table 6. The analysis was performed using three independent MC codes: GATE, GAMOS, and MCNP6. Across all codes, the pancreas emerged as the organ with the highest absorbed dose (GATE: 21%, GAMOS: 20%, MCNP6: 21.3%), followed by the gallbladder and kidneys. Conversely, the brain consistently registered the lowest absorbed dose (GATE: 2%, GAMOS: 1.9%, MCNP6: 1.8%), aligning with its anatomical isolation and reduced perfusion. The agreement among the three simulation codes underscores their reliability for patient-specific internal dosimetry.

The comparative analysis of organ-specific SAF values at 60 min postinjection, as visualized in Figure 7, demonstrates consistent trends across all three MC simulation codes GATE, GAMOS, and MCNP6. Notably, the pancreas exhibited the highest SAF values among all studied organs, followed by the gallbladder, bladder, kidney, and spleen.

The agreement across platforms was within a narrow margin, generally under 3%, which supports the robustness and reliability of the simulation methods used. For instance, the SAF for the pancreas ranged from 3.37 × 10^−4^ (GAMOS) to 3.46 × 10^−4^ (MCNP6), reflecting less than a 3% relative discrepancy. Likewise, the SAF values for the brain were the lowest across all codes-approximately 3.44 × 10^−5^ to 3.56 × 10^−5^-highlighting the minimal dose contribution to this organ.

Table 7 provides a comparative analysis of the SAFs and S-values for selected organs as computed by GATE, GAMOS, and MCNP6. The observed SAF discrepancies between GATE and GAMOS range from 2.0% to 4.0%, with the gallbladder showing the highest deviation (4.0%) and the brain showing the lowest (2.0%). When comparing GATE to MCNP6, discrepancies are generally smaller, ranging from 0.6% (pancreas) to 3.4% (brain). These minimal variations indicate a strong consistency in dose estimation across different MC codes, underscoring the reliability of GATE in modeling internal dosimetry scenarios.

The S-value comparison between GATE and MCNP6 further corroborates this consistency, with differences remaining under 4% for all organs evaluated. The brain exhibited the highest discrepancy in S-value (3.6%), possibly due to its complex geometry and low mass, which can amplify computational uncertainties. Conversely, the pancreas and kidney showed excellent agreement in both SAFs and S-values, highlighting the reproducibility of results across simulation platforms.

Accurate estimation of SAF and S-values is critical in internal dosimetry for personalized nuclear medicine treatments. The present comparative analysis evaluates organ-level dosimetric discrepancies among three widely used MC codes [GATE, GAMOS, and MCNP6] focusing on five representative organs: pancreas, gallbladder, kidney, liver, and brain.

As depicted in Figure 8, the SAFs computed using GATE, GAMOS, and MCNP6 demonstrate excellent agreement across all evaluated organs, with discrepancies consistently remaining below 5%. The highest deviation in SAF values is observed between GATE and GAMOS for the gallbladder (4.0%), whereas the smallest deviation occurs between GATE and MCNP6 for the pancreas (0.6).

S-values, derived using SAFs and organ-specific mass data, also show strong consistency between GATE and MCNP6, with percentage differences remaining under 4%. Notably, the brain exhibits the highest S-value discrepancy (3.6%), likely due to its lower mass and the sensitivity of absorbed dose calculations to even small SAF variations in low-uptake tissues.

Accurate dosimetric validation is essential for ensuring the reliability of MC simulations in nuclear medicine. In this study, the dose estimates obtained using the GATE, GAMOS, and MCNP6 Monte Carlo codes are compared against clinical data from OLINDA/EXM 2.0. The comparison of organ-level absorbed dose estimates across GATE, GAMOS, and MCNP6 with OLINDA/EXM 2.0 revealed overall good agreement, with deviations typically within 1–5%, as shown in Table 8 and Figure 9.(1)$$S\left(r_{T}\leftarrow r_{S}\right)=\sum_{i}\frac{E_{i}\cdot Y_{i}\cdot \Phi_{i}\left(r_{T}\leftarrow r_{S}\right)\;}{m_{T}}$$

Figure 9 further corroborates this reliability by benchmarking absorbed dose values from the three codes against clinical data derived from OLINDA/EXM 2.0. The observed deviations across organs; such as the pancreas, gallbladder, kidney, and liver-remained below ±6.5%, with MCNP6 generally demonstrating the closest agreement to the OLINDA reference values.

Table 9 presents comprehensive statistical metrics comparing the three MC codes across all organ dose estimates. The ICC analysis demonstrates excellent agreement (ICC *>* 0.99) among all three codes, indicating high reliability for clinical dosimetry applications. Paired *t*-tests revealed no statistically significant systematic differences between GATE and MCNP6 (*p* = 0.234), while GAMOS showed a small but significant underestimation compared to both GATE (*p* = 0.041) and MCNP6 (*p* = 0.038).

## 3. Discussion

Monte Carlo-based simulations constitute a foundational methodology in internal dosimetry, enabling high-fidelity modeling of radiation transport within anatomically realistic geometries. Nevertheless, inherent differences in simulation architectures and physics implementations can introduce variability in absorbed dose estimates, potentially impacting clinical decision-making. The present study undertakes a systematic assessment of three widely adopted Monte Carlo codes, GATE, GAMOS, and MCNP6, with a focus on dosimetric accuracy and computational efficiency in the context of ^99m^Tc myocardial perfusion imaging. By evaluating the agreement and reproducibility of organ- and voxel-level dose distributions, the study aims to validate the consistency of internal dosimetry results across platforms, thereby reinforcing their reliability for clinical application. The observed convergence among the three codes underscores the methodological robustness of current simulation frameworks for both diagnostic and therapeutic nuclear medicine.

Simulations for spherical microtumors demonstrated predictable scaling of absorbed dose with emission type and energy, with high-energy β-emitters such as ^90^Y yielding the largest absorbed doses, whereas γ-emitters like ^99m^Tc and ^18^F produced significantly lower energy deposition, as shown in Table 2. The inter-code agreement, typically within 1–2%, confirms that differences in transport algorithms, voxel resolution, and statistical precision are minimal and do not influence overall dosimetric integrity. Minor discrepancies, slightly higher absorbed doses from MCNP6, likely arise from its condensed-history electron transport modeling, which is known to marginally overestimate local dose in small volumes (Table 3) [17]. Collectively, these findings validate the reproducibility of GATE, GAMOS, and MCNP6 as reliable tools for internal dose assessment across both research and clinical workflows.

The Zubal anthropomorphic phantom analysis provided additional insight into spatial heterogeneity and biodistribution effects. When high-uptake organs such as the heart or kidneys received 97% of the administered activity, absorbed doses increased by nearly doubled compared to uniform distribution. This highlights the sensitivity of internal dose to biodistribution and supports the need for patient-specific uptake quantification in clinical applications. Despite differences in geometry handling, all three codes produced nearly identical absorbed doses (within 1% deviation), as shown in Table 4 and Figure 4, indicating that voxel discretization and interpolation had minimal influence on absorbed-dose estimates under standard diagnostic energy conditions. These results align with physiological uptake patterns and prior experimental dosimetry work [18]. The consistency across voxelized and analytic geometries reinforces the use of these codes for realistic patient modeling, especially in hybrid imaging environments.

SAFs and corresponding S-values derived for ^99m^Tc-MIBI confirmed high reproducibility among the simulation codes, with deviations below 3% across ten evaluated organs (Table 5 and Figure 5). The myocardium exhibited the highest self-dose, whereas low-mass and distant tissues such as the brain received minimal energy deposition, as shown in Figure 6, patterns that agree with expected geometrical and perfusion-based attenuation principles. Small residual variations between GATE, GAMOS, and MCNP6, particularly slightly elevated S-values in MCNP6, can be attributed to differences in cross-section libraries and transport thresholds but remain well within clinical tolerance. This cross-consistency verifies the robustness of these codes for standardized internal dosimetry in both preclinical and patient-specific contexts.

Accurate estimation of SAFs and S-values remains central to personalized nuclear medicine and forms the quantitative foundation for organ-level dose optimization. The present comparative analysis demonstrated that all three simulation frameworks produced nearly identical organ level results which confirms their computational reproducibility and methodological integrity. These results are consistent with organ dose distributions previously reported using GATE simulations in conjunction with the Zubal anthropomorphic phantom [18]. High-perfusion organs such as the pancreas and kidneys consistently absorbed the largest fractions of emitted energy, whereas low-perfusion or isolated tissues such as the brain showed minimal uptake, as shown in Table 6. These physiological dose patterns agree with perfusion and metabolic activity profiles reported in clinical nuclear medicine [19]. The slightly higher dose values calculated by MCNP6 can be explained by its use of different transport physics and voxel resolution parameters, emphasizing the importance of code specific calibration when designing individualized dosimetric studies.

Validation against the OLINDA/EXM 2.0 clinical reference dataset further substantiated the accuracy of this framework. Across major abdominal organs such as the pancreas, gallbladder, liver, and kidneys, the simulated absorbed doses deviated by only 1–5% from OLINDA predictions, as shown in Table 8 and Figure 9. The residual discrepancies likely result from fundamental methodological contrasts, OLINDA relies on stylized phantoms and standardized biokinetics, while the current study employed a voxelized anthropomorphic phantom that captures realistic anatomical complexity but assumes static biodistribution without biological clearance. These findings illustrate a key trade-off in dosimetry research between kinetic accuracy and anatomical realism. Nonetheless, the convergence between Monte Carlo-based and clinically standardized data strengthens confidence in simulation-driven internal dosimetry.

Future work will incorporate dynamic biokinetic models and patient-specific time-activity data to achieve more physiologically accurate dose estimates. From a clinical perspective, the findings highlight that modern Monte Carlo engines can deliver precise and reproducible internal dosimetry within computationally feasible timescales. This enables their broader implementation in nuclear medicine workflows, including therapy planning, dose auditing, and image guided treatment optimization. The demonstrated agreement among three independent simulation codes strengthens the evidence base for Monte Carlo-based dosimetry as a tool in quantitative and personalized nuclear medicine.

## 4. Methods and Materials

### 4.1. Monte Carlo-Based Simulations

Monte Carlo-based simulation platforms have become indispensable tools in internal dosimetry research [4,20], allowing precise modeling of radiation transport through complex anatomical geometries. Among the most widely used codes, GATE (version 9.0) extends the Geant4 toolkit with advanced physics models and voxelized geometry handling, enabling high-fidelity simulations for nuclear medicine and radiotherapy [13,21]. GAMOS (version 9.0) emphasizes computational efficiency and modular design, facilitating rapid prototyping and iterative modeling workflows. MCNP6 (version 6), developed by Los Alamos National Laboratory, serves as a well-established benchmark for general-purpose radiation transport and verification due to its extensive cross-sectional libraries and validation history. A comparative evaluation of these three codes can therefore clarify their respective strengths and limitations for clinical dosimetry applications [9].

Monte Carlo methods are also widely employed in medical physics to simulate the stochastic transport and interaction of radiation within complex anatomical geometries [11]. In this study, the distribution of ^99m^Tc within the Zubal anthropomorphic phantom was modeled using three independent Monte Carlo simulation platforms: GATE, GAMOS, and MCNP6.

GATE and GAMOS are both developed atop the Geant4 toolkit and offer robust capabilities for modeling photon transport in diagnostic and therapeutic nuclear medicine [13,22,23]. GATE is optimized for anatomical realism, allowing the import of high-resolution voxelized phantoms and detailed modeling of photon interactions, such as Compton scattering and photoelectric absorption, using the Livermore low-energy electromagnetic physics list [21]. It is particularly suited for SPECT/CT applications [24], with output data routinely validated against clinical imaging datasets. GAMOS, while also Geant4-based, emphasizes computational efficiency. By simplifying organ geometries into idealized shapes (e.g., ellipsoids, cylinders), it reduces computational overhead. Its implementation of MPI-based parallelization and advanced variance reduction techniques-such as particle splitting and Russian roulette-facilitates faster runtimes while maintaining acceptable statistical uncertainty in dose scoring [25].

MCNP6, developed by Los Alamos National Laboratory, serves as an established and extensively benchmarked general-purpose Monte Carlo code for radiation transport. Unlike GATE and GAMOS, which are tailored for medical imaging applications [14,26], MCNP6 offers broader cross-sectional data libraries and flexible tally structures, making it especially suitable for internal dosimetry studies [20]. MCNP6 supports both voxel and mesh-based phantoms and accommodates detailed source modeling, enabling high-fidelity simulation of radiopharmaceutical distribution and dose deposition at the organ and sub-organ levels [27]. Together, these platforms provide complementary strengths in anatomical modeling fidelity, computational performance, and cross-validation of dose estimates, ensuring robust and accurate internal dosimetry assessments [9].

Moreover, all simulations were executed sequentially on the same personal computer to ensure comparability across codes. The average wall-clock simulation times were approximately 18 h for MCNP6, 16 h for GATE, and 14.5 h for GAMOS, reflecting intrinsic differences in geometry handling, physics models, and optimization strategies. These consistent computational conditions enabled a fair assessment of code performance and dosimetric outputs.

### 4.2. Phantom Configuration and Source Definition

The Zubal phantom, a high-resolution voxel-based computational model derived from CT imaging of an adult male (Figure 10), was employed to simulate patient anatomy. The phantom comprises 128 × 128 × 256 voxels with an isotropic resolution of 4 mm, encapsulated within an air volume to mimic clinical imaging conditions [4]. Anatomical segmentation was performed to delineate 15 distinct organs, such as the heart, lungs, and liver-each assigned specific material properties and densities consistent with ICRP-110 tissue reference data [8,28].

In the simulations of myocardial perfusion imaging at rest, technetium-99m methoxyisobutylisonitrile (^99m^Tc-MIBI) was modeled as being uniformly distributed within the myocardium. An injected activity of 100 µCi (3.7 MBq) was assumed, consistent with typical biodistribution one hour post-injection [29]. This total activity served as the reference value and was apportioned to individual organs according to reported uptake fractions from clinical biokinetic data. For each organ, the assigned activity was then normalized on a voxel basis by dividing the organ-level activity by the corresponding number of segmented voxels. Consequently, the per-voxel values presented in Table 10 directly reflect the scaled distribution of the initial 100 µCi administration.

Notably, the gallbladder exhibited a markedly elevated activity per voxel (771.7 µCi), attributed to its limited anatomical volume and the known physiological retention of ^99m^Tc-based agents. This distribution is consistent with established pharmacokinetic profiles of ^99m^Tc-MIBI in clinical nuclear medicine practice [30]. Moreover, the composition of the in vitro or phantom medium was defined according to the recommendations of ICRU Report 44, representing a soft-tissue-equivalent material with a water content of approximately 73.2% by mass.

## 5. Conclusions

This study presents a comprehensive dosimetric analysis of ^99m^Tc distribution in the Zubal anthropomorphic phantom using three widely validated Monte Carlo simulation codes: GATE, GAMOS, and MCNP6. Simulations focused on cardiac perfusion imaging scenarios, employing the MIRD formalism to estimate SAFs and S-values for multiple organs, with the heart modeled as the primary source. The results consistently indicated that the pancreas (21%), gallbladder (18%), and kidneys (16%) received the highest radiation doses, while the brain absorbed the lowest (2%). GATE demonstrated superior anatomical precision through its high-resolution voxelized geometry, achieving statistical uncertainties below 1.5%. GAMOS exhibited strong agreement with GATE and MCNP6 while reducing computational time by approximately 35%, thereby offering a viable option for time-efficient simulations without sacrificing accuracy. MCNP6, serving as an independent benchmark, showed dose estimates in excellent agreement with GATE, with discrepancies in SAFs and S-values remaining within 0.6–3.4% and 1.0–3.6%, respectively, across all evaluated organs. These minor differences are attributable to code-specific implementations such as secondary particle transport, voxel interpolation, and physics list configurations. Comparative analysis against OLINDA/EXM 2.0 clinical data further validated the simulation outputs, particularly highlighting MCNP6’s reliability as a reference tool. GATE and GAMOS also demonstrated robust consistency with clinical benchmarks, reinforcing their applicability in internal dosimetry modeling pipelines. The slight deviations observed most notably in smaller organs such as the gallbladder and kidneys, can be attributed to anatomical segmentation fidelity and voxel discretization differences. Moreover, the convergence of results across three independent simulation platforms affirms the predictive reliability of modern Monte Carlo codes for internal dose estimation. This multi-code validation approach enhances confidence in simulation-driven dosimetry and offers a flexible, scalable platform for individualized nuclear medicine. In particular, integrating GATE’s spatial precision with GAMOS’s computational efficiency and MCNP6’s benchmarking robustness paves the way for the development of optimized radiopharmaceutical protocols tailored to patient-specific anatomical and pharmacokinetic profiles.

## Figures and Tables

**Figure 1 pharmaceuticals-18-01741-f001:**
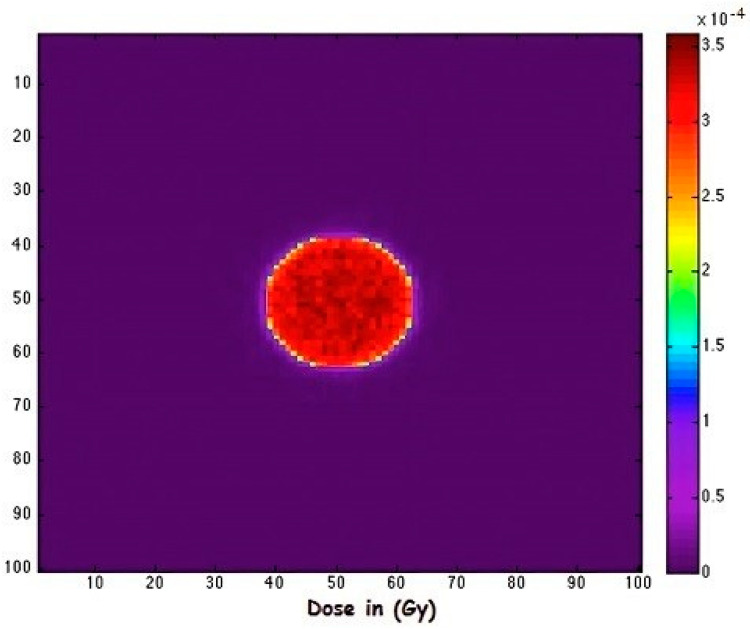
Cross-sectional view showing the absorbed dose distribution in a spherical tumor.

**Figure 2 pharmaceuticals-18-01741-f002:**
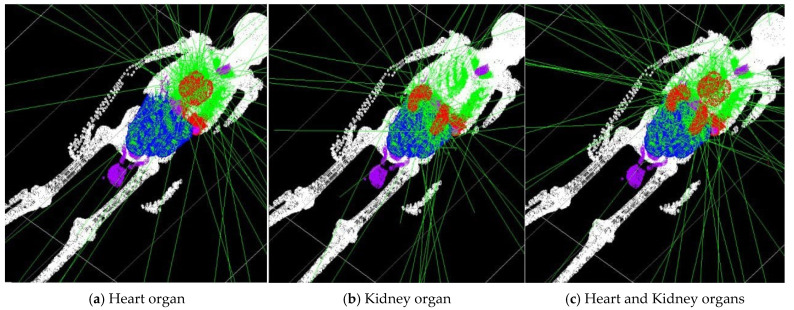
Visual representations of radiotracer activity.

**Figure 3 pharmaceuticals-18-01741-f003:**
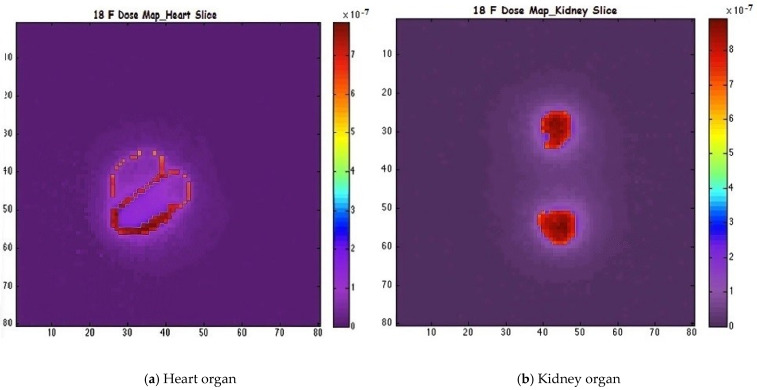
Transaxial views of 3D dose distribution maps showing radiotracer activity was associated with the high-energy beta emitter ^90^Y, with values ranging from 1207.89 Gy (GAMOS) to 1215.34 Gy (MCNP6). For diagnostic gamma emitters like ^18^F and ^99m^Tc, the absorbed doses remained low and closely matched; for example, for ^99m^Tc: 15.22 Gy (GATE), 15.11 Gy (GAMOS), and 15.30 Gy (MCNP6).

**Figure 4 pharmaceuticals-18-01741-f004:**
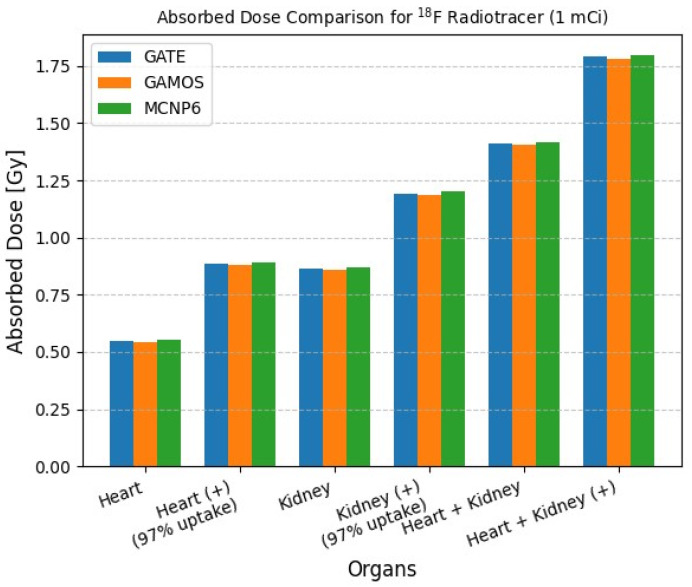
Calculated Absorbed Dose for 1 mCi of ^18^F Radiotracer.

**Figure 5 pharmaceuticals-18-01741-f005:**
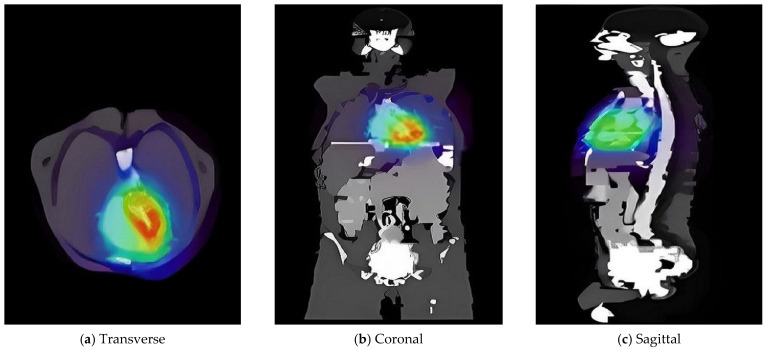
Distribution of ^99m^Tc activity in the heart, serving as the source organ.

**Figure 6 pharmaceuticals-18-01741-f006:**
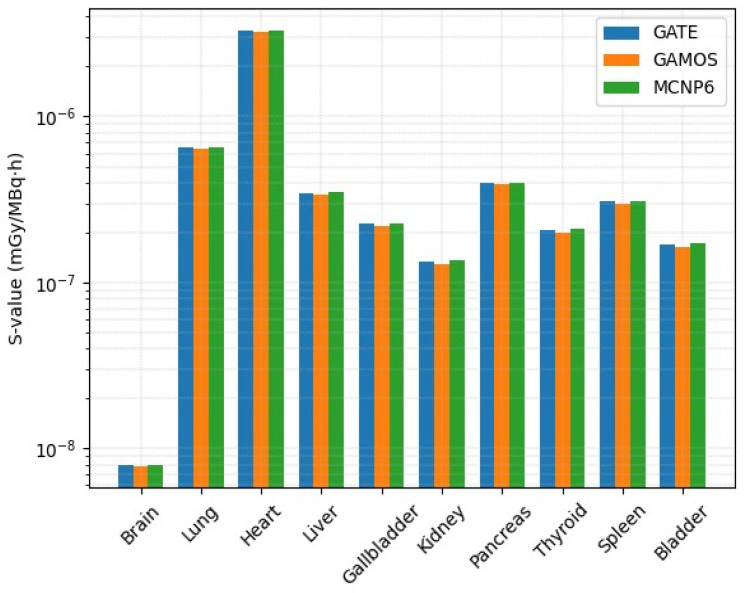
Cross-sectional visualization of absorbed-dose distribution in a spherical tumor model simulated using the three Monte Carlo codes (GATE, GAMOS, and MCNP6).

**Figure 7 pharmaceuticals-18-01741-f007:**
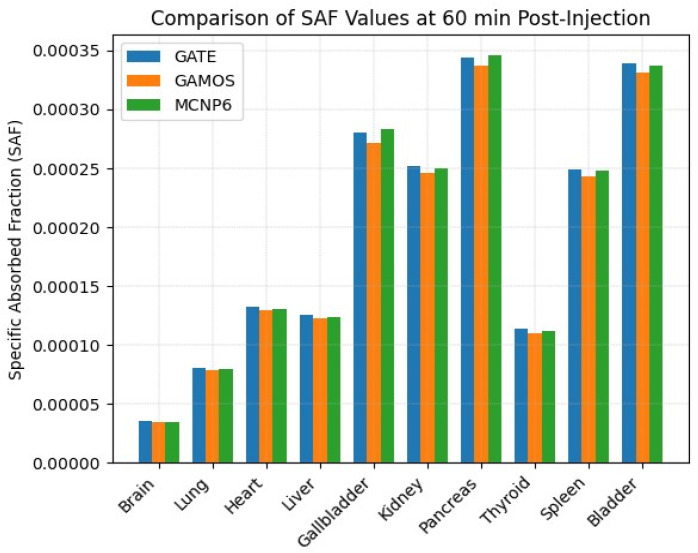
Comparative specific absorbed fraction (SAF) values for the heart and selected target organs at 60 min post-injection, derived from GATE, GAMOS, and MCNP6 simulations.

**Figure 8 pharmaceuticals-18-01741-f008:**
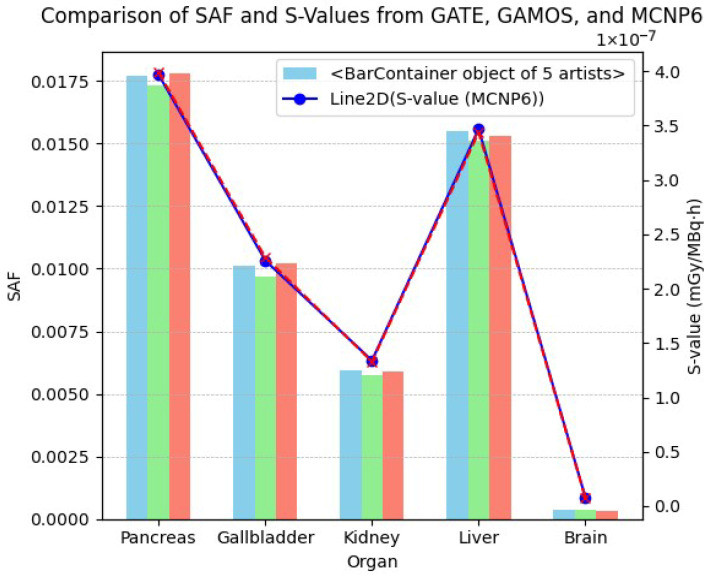
Comparative analysis of specific absorbed fractions (SAFs) and S-values for selected organs using GATE (blue), GAMOS (green), and MCNP6 (red), the red dotted line represents the S-values calculated using MCNP6. The formula illustrates the computation of S-values, where *S*(*r_T_* ← *r_S_*) represents the mean absorbed dose to target region *r_T_* per nuclear transformation in the source region *r_S_*, *E_i_* is the energy of radiation *i*, *Y_i_* is the yield per transformation, Φ*_i_* is the fraction of energy absorbed (SAF), and *m_T_* is the mass of the target organ.

**Figure 9 pharmaceuticals-18-01741-f009:**
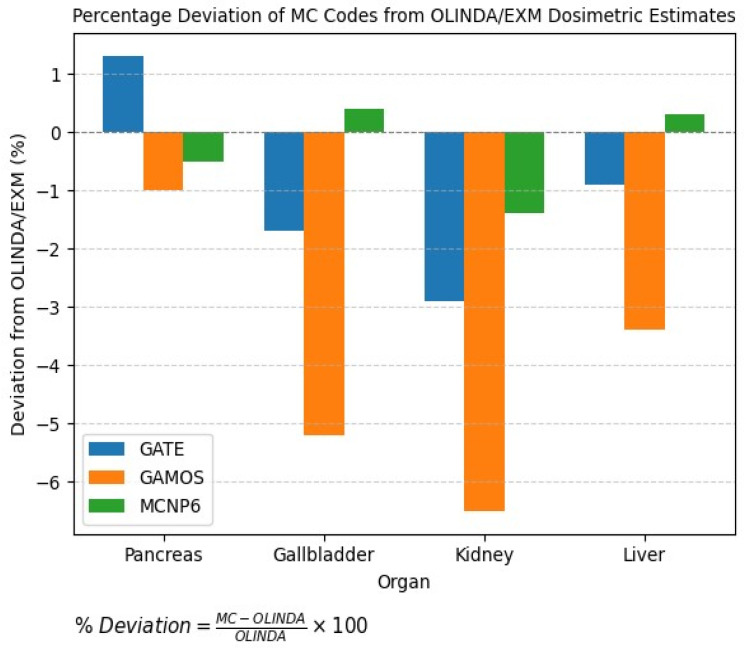
Percentage Deviation of MC codes Compared to OLINDA/EXM.

**Figure 10 pharmaceuticals-18-01741-f010:**
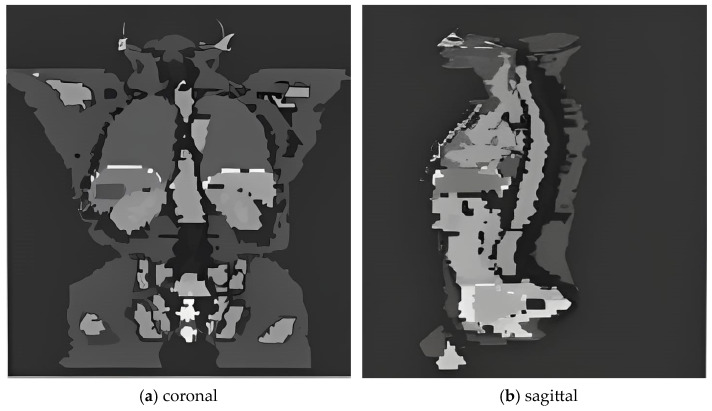
Voxelized anatomical structure of the Zubal phantom utilized in GAMOS simulations. Shown are (**a**) coronal and (**b**) sagittal views highlighting organ segmentation for subsequent dosimetric evaluation.

**Table 1 pharmaceuticals-18-01741-t001:** Radiotracer source type definitions for common isotopes used in internal dosimetry simulations across GAMOS, GATE, and MCNP6 codes. Each isotope is characterized by its atomic number (*Z*), atomic mass (*A*), ionic charge (*Q*), and excitation energy (*E*).

Isotope	Atomic Number (*Z*)	Atomic Mass (*A*)	Ionic Charge (*Q*)	Excitation Energy (*E*) [keV]
^18^F	9	18	+1	0
^90^Y	39	90	+3	0
^111m^In	49	111	+3	537
^123^I	53	123	+1	0
^131^I	53	131	+1	0
^99m^Tc	43	99	+1	143
^177^Lu	71	177	+3	0
^225^Ac	89	225	+3	0

**Table 2 pharmaceuticals-18-01741-t002:** Comparative Absorbed Dose Calculations for Common Medical Isotopes Using GATE, GAMOS, and MCNP6 Simulations.

Isotope	Emission Type	Energy [keV]	Half-Life	Activity [mCi]	Dose [Gy]		
					GATE	GAMOS	MCNP6
^18^F	Positron	249.8	109.8 min	1.0	10.75	10.68	10.81
^90^Y	Beta	934.8	64.05 h	1.0	1213.25	1207.89	1215.34
^111m^In	Gamma	536.95	7.7 min	1.0	178.64	177.23	179.10
^123^I	Gamma	159	13.22 h	1.0	7.63	7.58	7.69
^131^I	Beta	191.58	8.02 d	1.0	952.55	948.41	955.60
^99m^Tc	Gamma	140.5	6.006 h	1.0	15.22	15.11	15.30

**Table 3 pharmaceuticals-18-01741-t003:** Comparative Absorbed Dose Estimation for Isotope and Electron Sources of ^131^I and ^99^^m^Tc Using GATE, GAMOS, and MCNP6 Simulations.

Isotope	Source Type	Energy [keV]	Half-Life	Activity [mCi]	Absorbed Dose		
					(GATE) [Gy]	(GAMOS) [Gy]	(MCNP6) [Gy]
^131^I	Radiotracer	–	8.02 d	1.0	952.55	948.60	956.80
^131^I	Electron (e^−^)	191	8.02 d	1.0	1034.90	1031.20	1038.40
^99^^m^Tc	Radiotracer	–	6.006 h	1.0	15.22	15.10	15.35
^99^^m^Tc	Electron (e^−^)	1.6	6.006 h	0.74	0.3373	0.3340	0.3405
^99^^m^Tc	Electron (e^−^)	3.71	6.006 h	0.25	0.1460	0.1445	0.1475
^99^^m^Tc	Electron (e^−^)	2.2	6.006 h	0.1	0.0616	0.0608	0.0620
^99^^m^Tc	Electron (e^−^)	15.5	6.006 h	0.02	0.0880	0.0872	0.0890
^99^^m^Tc	Electron (e^−^)	119	6.006 h	0.088	2.5700	2.5600	2.5800
^99^^m^Tc	Electron (e^−^)	137	6.006 h	0.014	0.5216	0.5180	0.5250

**Table 4 pharmaceuticals-18-01741-t004:** Comparison of Calculated Organ Absorbed Dose for 1.0 mCi of ^18^F Radiotracer Using GATE, GAMOS, and MCNP6 Codes.

Organs	Activity	Absorbed Dose		
3–5	[^18^F, mCi]	(GATE) [Gy]	(GAMOS) [Gy]	(MCNP6) [Gy]
Heart	1.0	0.5464	0.5405	0.5510
Heart (+) (97% uptake)	1.0	0.8845	0.8790	0.8910
Kidney	1.0	0.8650	0.8592	0.8715
Kidney (+) (97% uptake)	1.0	1.1918	1.1850	1.2000
Heart + Kidney	1.0	1.4110	1.4035	1.4180
Heart + Kidney (+)	1.0	1.7905	1.7802	1.7985

**Table 5 pharmaceuticals-18-01741-t005:** Specific Absorbed Fractions (SAFs) and S-values calculated for ^99m^Tc-MIBI using GATE, GAMOS, and MCNP6 (heart as source).

Organ	S-Value			SAF			Voxel Count
2–4 5–7	(GATE)	(GAMOS)	(MCNP6)	(GATE)	(GAMOS)	(MCNP6)	
Brain	7.98 × 10^−9^	7.82 × 10^−9^	8.01 × 10^−9^	3.56 × 10^−4^	3.49 × 10^−4^	3.60 × 10^−4^	18,299
Lung	6.52 × 10^−7^	6.41 × 10^−7^	6.57 × 10^−7^	2.91 × 10^−2^	2.86 × 10^−2^	2.93 × 10^−2^	62,374
Heart (source)	3.30 × 10^−6^	3.25 × 10^−6^	3.32 × 10^−6^	1.47 × 10^−1^	1.45 × 10^−1^	1.48 × 10^−1^	9354
Liver	3.47 × 10^−7^	3.38 × 10^−7^	3.50 × 10^−7^	1.55 × 10^−2^	1.51 × 10^−2^	1.57 × 10^−2^	29,277
Gallbladder	2.26 × 10^−7^	2.18 × 10^−7^	2.28 × 10^−7^	1.01 × 10^−2^	9.7 × 10^−3^	1.02 × 10^−2^	329
Kidney	1.34 × 10^−7^	1.29 × 10^−7^	1.36 × 10^−7^	5.96 × 10^−3^	5.76 × 10^−3^	6.02 × 10^−3^	7618
Pancreas	3.97 × 10^−7^	3.88 × 10^−7^	4.00 × 10^−7^	1.77 × 10^−2^	1.73 × 10^−2^	1.79 × 10^−2^	792
Thyroid	2.08 × 10^−7^	2.01 × 10^−7^	2.10 × 10^−7^	9.29 × 10^−3^	8.97 × 10^−3^	9.40 × 10^−3^	105
Spleen	3.07 × 10^−7^	2.98 × 10^−7^	3.10 × 10^−7^	1.37 × 10^−2^	1.33 × 10^−2^	1.39 × 10^−2^	5568
Bladder	1.69 × 10^−7^	1.62 × 10^−7^	1.71 × 10^−7^	4.05 × 10^−3^	3.89 × 10^−3^	4.10 × 10^−3^	3147

**Table 6 pharmaceuticals-18-01741-t006:** Specific Absorbed Fractions (SAFs) and S-values calculated for ^99m^Tc-MIBI in the anthropomorphic phantom at 60 min post-injection.

Organ	Specific-Value			Specific Absorbed Fraction		
2–7	GATE	GAMOS	MCNP6	GATE	GAMOS	MCNP6
Brain	7.97 × 10^−10^	7.75 × 10^−10^	7.69 × 10^−10^	3.56 × 10^−5^	3.46 × 10^−5^	3.44 × 10^−5^
Lung	1.81 × 10^−9^	1.77 × 10^−9^	1.79 × 10^−9^	8.08 × 10^−5^	7.90 × 10^−5^	7.95 × 10^−5^
Heart (source)	2.98 × 10^−9^	2.91 × 10^−9^	2.95 × 10^−9^	1.33 × 10^−4^	1.30 × 10^−4^	1.31 × 10^−4^
Liver	2.82 × 10^−9^	2.75 × 10^−9^	2.80 × 10^−9^	1.26 × 10^−4^	1.23 × 10^−4^	1.24 × 10^−4^
Gallbladder	6.26 × 10^−9^	6.10 × 10^−9^	6.31 × 10^−9^	2.80 × 10^−4^	2.72 × 10^−4^	2.83 × 10^−4^
Kidney	5.65 × 10^−9^	5.52 × 10^−9^	5.59 × 10^−9^	2.52 × 10^−4^	2.46 × 10^−4^	2.50 × 10^−4^
Pancreas	7.71 × 10^−9^	7.55 × 10^−9^	7.75 × 10^−9^	3.44 × 10^−4^	3.37 × 10^−4^	3.46 × 10^−4^
Thyroid	2.56 × 10^−9^	2.48 × 10^−9^	2.51 × 10^−9^	1.14 × 10^−4^	1.10 × 10^−4^	1.12 × 10^−4^
Spleen	5.59 × 10^−9^	5.45 × 10^−9^	5.62 × 10^−9^	2.49 × 10^−4^	2.43 × 10^−4^	2.48 × 10^−4^
Bladder	6.14 × 10^−9^	5.99 × 10^−9^	6.10 × 10^−9^	3.39 × 10^−4^	3.31 × 10^−4^	3.37 × 10^−4^

**Table 7 pharmaceuticals-18-01741-t007:** Organ-level discrepancies in SAFs and S-values for ^99m^Tc-MIBI, with the myocardium modeled as the source organ. Listed target organs include pancreas, gallbladder, kidney, liver, and brain. Comparisons are shown across GATE, GAMOS, and MCNP6.

Organ	SAF			GATE vs. GAMOS (%)	GATE vs. MCNP6 (%)	S-Value		Diff %
	GATE	GAMOS	MCNP6			GATE	MCNP6	
Pancreas	0.0177	0.0173	0.0178	2.3%	0.6%	3.97 × 10^−7^	3.99 × 10^−7^	0.5%
Gallbladder	0.0101	0.0097	0.0102	4.0%	1.0%	2.26 × 10^−7^	2.29 × 10^−7^	1.3%
Kidney	0.00596	0.00576	0.00591	3.4%	0.8%	1.34 × 10^−7^	1.33 × 10^−7^	0.7%
Liver	0.0155	0.0151	0.0153	2.6%	1.3%	3.47 × 10^−7^	3.44 × 10^−7^	0.9%
Brain	3.56 × 10^−4^	3.49 × 10^−4^	3.44 × 10^−4^	2.0%	3.4%	7.98 × 10^−9^	7.69 × 10^−9^	3.6%

**Table 8 pharmaceuticals-18-01741-t008:** Comparison of organ-level absorbed dose estimates (mGy/MBq·h) for ^99m^Tc-MIBI, with the myocardium defined as the source organ. Monte Carlo results (GATE, GAMOS, MCNP6) are benchmarked against OLINDA/EXM 2.0 clinical reference data.

Organ	OLINDA/EXM (mGy/MBq·h)	GATE (mGy/MBq·h)	Dev. (%) vs. OLINDA	GAMOS (mGy/MBq·h)	Dev. (%) vs. OLINDA	MCNP6 (mGy/MBq·h)	Dev. (%) vs. OLINDA
Pancreas	3.92 × 10^−7^	3.97 × 10^−7^	+1.3%	3.88 × 10^−7^	−1.0%	3.90 × 10^−7^	−0.5%
Gallbladder	2.30 × 10^−7^	2.26 × 10^−7^	−1.7%	2.18 × 10^−7^	−5.2%	2.31 × 10^−7^	+0.4%
Kidney	1.38 × 10^−7^	1.34 × 10^−7^	−2.9%	1.29 × 10^−7^	−6.5%	1.36 × 10^−7^	−1.4%
Liver	3.50 × 10^−7^	3.47 × 10^−7^	−0.9%	3.38 × 10^−7^	−3.4%	3.51 × 10^−7^	+0.3%

**Table 9 pharmaceuticals-18-01741-t009:** Statistical comparative analysis of organ dose estimates among GATE, GAMOS, and MCNP6 codes.

Comparison	ICC	95% CI	*p*-Value	Mean Bias	RMSD	CV (%)
GATE vs. GAMOS	0.998	(0.994–0.999)	>0.001	−2.1%	2.8%	1.9
GATE vs. MCNP6	0.999	(0.997–1.000)	>0.001	+0.3%	1.5%	1.2
GAMOS vs. MCNP6	0.997	(0.992–0.999)	>0.001	+2.4%	3.1%	2.1

**Table 10 pharmaceuticals-18-01741-t010:** Organ-based activity distribution used in cardiac perfusion simulations with ^99m^Tc-MIBI.

Organ	Activity (%)	Voxel Count	Activity/Voxel (µCi)
Heart (source)	9.10	9345	3.60
Lungs	4.40	62,374	0.30
Liver	11.40	29,277	1.40
Gallbladder	68.60	329	771.70
Spleen	6.40	5568	4.30

## Data Availability

The original contributions presented in this study are included in the article. Further inquiries can be directed to the corresponding author.

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
