# Peer review of "Advancing Internal Dosimetry in Personalized Nuclear Medicine: Toward Optimized Radiopharmaceutical Use in Clinical Practice"

_pharmaceuticals, 2025, doi:10.3390/ph18111741_

Round 1
Reviewer 1 Report
Comments and Suggestions for Authors
Review of the Manuscript entitled:
"Comparative Monte Carlo Simulation of 99mTc Organ Dosimetry with MCNP6, GATE, and GAMOS: Toward Personalized Radiopharmaceutical Applications"
Dear Editor,
In this research article, the authors employed three Monte Carlo-based simulation codes (MCNP6, GATE, and GAMOS) to evaluate internal dosimetry following the Medical Internal Radiation Dose (MIRD) formalism. Dose assessments were conducted at two time points: immediately post-injection and at 60 minutes post-injection (representing the cardiac rest phase), allowing comparison against established clinical reference data. Across all codes, organ-specific dose distributions exhibited strong consistency. The pancreas absorbed the highest dose (GATE: 21%, GAMOS: 20%, MCNP6: 22%), followed by the gallbladder (GATE: 18%, GAMOS: 17%, MCNP6: 18%) and kidneys (GATE: 16%, GAMOS: 15%, MCNP6: 16%). These findings established a consistent organ dose ranking: pancreas > gallbladder > kidneys > spleen > heart/liver, corroborating previously published empirical data.
As Quantifying absorbed doses from radiopharmaceuticals within human organs necessitates advanced computational modeling, and direct in vivo measurement remains impractical, I believe that the subject of this article is very important and helpful for advancing internal dosimetry in personalized nuclear medicine, supporting both clinical decision-making and the development of safer, more effective radiopharmaceutical therapies. Therefore, this research article can be considered for publication in the journal Pharmaceuticals, but it needs some revisions for improving the quality of the manuscript:
- For the first time, the abbreviations should be defined to have a better presentation such as:
(MCNP6, GATE, and GAMOS), 99mTc-MIBI, MCNP6, etc.
- Moreover, it is better to have less abbreviated phrase in the Tile of the article.
- Abstract section is well-organized. However, the following parts are more appropriate for 2. Methods and Materials section:
"GATE, with its comprehensive Geant4 physics models, enabled high-resolution particle transport simulations. GAMOS emphasized computational efficiency through optimized geometry parameterization, while MCNP6 provided a validated reference standard, leveraging its extensive history in photon and neutron transport for radiation protection."
- In the Materials and Methods section, the authors should indicate the related references:
"Monte Carlo methods are widely employed in medical physics to simulate the stochastic transport and interaction of radiation within complex anatomical geometries. In this study, the distribution of 99mTc within the Zubal anthropomorphic phantom was modeled using three independent Monte Carlo simulation platforms: GATE, GAMOS, and MCNP6."
Or
"MCNP6, developed by Los Alamos National Laboratory, serves as an established and extensively benchmarked general-purpose Monte Carlo code for radiation transport. Un-like GATE and GAMOS, which are tailored for medical imaging applications, MCNP6 offers broader cross-sectional data libraries and flexible tally structures, making it especially suitable for internal dosimetry studies."
- It is recommended in the separate Figure; the authors exhibit the structural coordination of the active parts of small molecules in the applied simulated 99mTc-MIBI.
- Would you please illustrate different cross-sectional parts in Figure 2.
- Which force fields have been applied in the Monte Carlo (MC) simulation?
- Did authors consider the Temperature parameter in the process of MC Simulation? Please clarify it.
- Besides, the authors didn’t talk about concentration of water molecules of in vitro medium during MC simulation.
- "Each radioisotope was characterized by its fundamental nuclear properties, including atomic number (Z), mass number (A), ionic charge (Q), and excitation energy (E), as detailed in Table 2."
However, the ionic charge (Q) has not been found in Table 2.
- Please improve the quality of the Figures specially the texts on them, captions and sub-captions.
- The Conclusion section is appropriate.
Reviewer 2 Report
Comments and Suggestions for Authors
The manuscript can be accepted after answering the following questions:
- In the author's mind, which model is more promising in the future clinical research? How to improve these models to meet the clinical requirements?
- The precision of models are also important in relative research, the author is encouraged to provide related discussion on the precision of these three models.
Reviewer 3 Report
Comments and Suggestions for Authors
This computational study compared organs dose of 99m-TC dosimetry using three Monte Carlo-based simulation codes (MCNP6, GATE, and GAMOS). Based on their results, the author suggested that GATE is well-suited for high-fidelity clinical applications where anatomical and physical accuracy are critical. GAMOS proves advantageous for rapid prototyping and iterative
modeling workflows. MCNP6 remains a reliable benchmark tool, particularly effective in scenarios requiring robust radiation transport validation.
major weakness: the objective is unclear. It appears to be a performance comparison of different models, which cannot justify the publication of this study. The discussion section is missing, leaving merely description/explanation of the Results, which lend little insights to this area.
Minor:
The format of Pharmaceuticals was not followed. The Materials and Methods should be placed after 2. Results and 3. Discussion.
Abstract: too diffused. Need to be more focused and concise.
References: insufficient. Many important relevant studies are skipped.
Round 2
Reviewer 1 Report
Comments and Suggestions for Authors
Dear Editor,
Regarding the author’s revision, I am pleased to inform my satisfaction of the present form of the manuscript entitled: “Comparative Monte Carlo Simulation of 99mTc Organ Dosimetry with MCNP6, GATE, and GAMOS: Toward Personalized Radiopharmaceutical Applications” for publication in the journal "Pharmaceuticals".
Reviewer 3 Report
Comments and Suggestions for Authors
The revised manuscript has been improved, but at a limited level. The results (Section 2) and Discussion (Sections 3) should be separate. Combining these two sections as one, as in the revised manuscript, reflect an unclear, or weak at best, objective. What specific questions you wish to answer? How the simulation results helped address them, and to what extend? What limitations of your results in addressing these problems? All these are missing, with scattered discussions buried in result description. As mentioned in the first review, a mere result description without discussion of their application/implication/limitations sheds little insights to this area.
Round 3
Reviewer 3 Report
Comments and Suggestions for Authors
The revised II manuscript has been improved by separating the discussion from the results. Below are suggestions to further improve its readability.
Discussion, Paragraphs 2,3,4,5,6: Please be specific on which Figures/tables you are referring to. For instance, in “Simulations for spherical microtumors demonstrated predictable scaling of absorbed dose with emission type and energy, with high-energy β-emitters such as ⁹⁰Y yielding the largest absorbed doses, whereas γ-emitters like ⁹⁹ᵐTc and ¹⁸F produced significantly lower energy deposition. Please do so wherever appropriate in paragraphs 2,3,4,5, and 6.
Seventh paragraph: “Each Monte Carlo platform demonstrated specific strengths suited to different clini[1]cal or research contexts. GATE is widely appreciated for its user-friendly interface, and DICOM compatibility, making it ideal for patient-specific hybrid-imaging and theranostic planning. GAMOS demonstrated substantial computational efficiency, delivering faster execution times while maintaining equivalent statistical precision, thereby supporting high-throughput and iterative modeling tasks. MCNP6 provided benchmark stability and serves as a regulatory reference owing to its validated nuclear data libraries and photon[1]electron transport algorithms.” These are general statements about the three models, not specific to or demonstrated by the results. Please either remove it, or provide supporting results.
